# Genetic Dissection of Drought Tolerance in Maize Through GWAS of Agronomic Traits, Stress Tolerance Indices, and Phenotypic Plasticity

**DOI:** 10.3390/ijms26136285

**Published:** 2025-06-29

**Authors:** Ronglan Li, Dongdong Li, Yuhang Guo, Yueli Wang, Yufeng Zhang, Le Li, Xiaosong Yang, Shaojiang Chen, Tobias Würschum, Wenxin Liu

**Affiliations:** 1Sanya Institute of China Agricultural University, China Agricultural University, Sanya 572025, China; 2State Key Laboratory of Maize Bio-Breeding, National Maize Improvement Center, College of Agronomy and Biotechnology, China Agricultural University, Beijing 100193, China; 3Institute of Plant Breeding, Seed Science and Population Genetics, University of Hohenheim, 70599 Stuttgart, Germany

**Keywords:** maize, drought tolerance, genome-wide association study, agronomic traits, stress tolerance indices, phenotypic plasticity

## Abstract

Drought severely limits crop yield every year, making it critical to clarify the genetic basis of drought tolerance for breeding of improved varieties. As drought tolerance is a complex quantitative trait, we analyzed three phenotypic groups: (1) agronomic traits under well-watered (WW) and water-deficit (WD) conditions, (2) stress tolerance indices of these traits, and (3) phenotypic plasticity, using a multi-parent doubled haploid (DH) population assessed in multi-environment trials. Genome-wide association studies (GWAS) identified 130, 171, and 71 quantitative trait loci (QTL) for the three groups of phenotypes, respectively. Only one QTL was shared among all trait groups, 25 between stress indices and agronomic traits, while the majority of QTL were specific to their group. Functional annotation of candidate genes revealed distinct pathways of the three phenotypic groups. Candidate genes under WD conditions were enriched for stress response and epigenetic regulation, while under WW conditions for protein synthesis and transport, RNA metabolism, and developmental regulation. Stress tolerance indices were enriched for transport of amino/organic acids, epigenetic regulation, and stress response, whereas plasticity showed enrichment for environmental adaptability. Transcriptome analysis of 26 potential candidate genes showed tissue-specific drought responses in leaves, ears, and tassels. Collectively, these results indicated both shared and independent genetic mechanisms underlying drought tolerance, providing novel insights into the complex phenotypes related to drought tolerance and guiding further strategies for molecular breeding in maize.

## 1. Introduction

Maize (*Zea mays* L.) is one of the most important crops worldwide, being used as resource for food, feed and biofuel [1,2,3]. The wide demand of maize makes grain yield a major breeding goal. However, with global climate change and water shortage, drought stress constitutes one of the major constraints to agriculture and causes serious crop yield losses worldwide [4]. In maize, the most critical stages sensitive to drought stress are the pre-flowering and flowering stages [5], followed by early vegetative stages and grain filling [6,7]. Drought stress can severely affect the growth and development of maize, for example, decreasing photosynthesis [8], leading to leaf wilting [9], decreased plant height [10], delayed silking and increased anthesis-silking interval [11], decreased pollination and seed setting rate, and consequently reducing maize yield [12,13]. In particular, a prolonged drought during flowering and grain filling can lead to high yield losses [14]. Thus, it is necessary to explore the genetic mechanisms underlying drought tolerance in order to develop new drought-tolerant varieties.

Dissecting the genetic basis of drought tolerance is crucial for molecular breeding, but drought tolerance is a highly quantitative trait. In the last decade, most QTL mapping studies adopted the strategy to measure important agronomic traits under drought and under well-watered conditions, revealing a large number of QTL [15,16,17,18,19,20]. In addition, some of the studies developed stress tolerance indices for agronomic traits and selection of drought-tolerant genotypes by calculating the relative difference or the ratio between water-deficit and well-watered conditions [21]. Among the more than ten stress tolerance indices developed to date [22,23], the stress tolerance index (STI) and the drought resistance index (DRI) were shown to be stable and reliable [24,25,26,27]. In addition, most studies are based on genotypic effect (G) for mapping, but only few have investigated genotype-by-environment interaction (G×E) patterns. The identification of G×E interaction and its genetic basis in multi-environmental trials can deepen the understanding of the genetic response to drought [28]. Phenotypic variation among genotypes in different environments is assessed as the degree of G×E interaction, which is also known as phenotypic plasticity [29]. Linear plasticity (LP) of phenotypic plasticity can be quantified through the regression model proposed by Finlay and Wilkinson [30] and some studies have explored the genetic architecture of phenotypic plasticity by LP in crops [31,32,33]. Notably, diverse phenotypes can reveal hidden drought tolerance candidate genes. Previous studies have explored the genetic mechanisms and identified candidate genes associated with agronomic traits under different water conditions, stress tolerance indices, and phenotypic plasticity. However, almost no research has integrated these three aspects to explore their shared and unique genetic architectures. This study therefore aims to dissect the common and distinct genetic mechanisms underlying these three trait groups, thereby facilitating breeders to efficiently develop high-yielding maize varieties with superior drought tolerance and adaptability.

Genome-wide association study (GWAS) is a powerful tool for detecting genomic polymorphisms associated with complex traits [34]. GWAS have successfully identified many drought tolerance QTL and even genes in maize, such as *ZmDnaJ*–*ZmNCED6* module [35], *ZmNAC080308* [36], *ZmcPGM2* and *ZmFAB1A* [37], *ZmEXPA4* [38], and *ZmVPP1* [39]. GWAS are commonly performed with diversity panels, but multi-parental populations are an attractive alternative as they offer a higher QTL detection power due to the more balanced allele frequencies and a lower rate of false positive due to better controlled population structure [40]. Thus, multi-parental populations can effectively combine the advantages of biparental populations used for linkage mapping and diversity panels used for association mapping and overcome their shortcomings. A large number of studies have been published using different types of such multi-parental populations, such as the nested association mapping (NAM) population [41,42,43], multiparent advanced generation intercross (MAGIC) population [44,45], and random-open-parent association mapping (ROAM) population [46,47]. These multi-parental populations still require a longer period of time to establish. In contrast, multi-parental double haploid (DH) populations can greatly shorten the breeding time. The DH technology has become an efficient alternative in maize breeding as it allows to generate completely homozygous lines in just two generations [48]. Multi-parental DH populations were used for mapping of flowering time [49] and tolerance to low phosphorus in maize [32].

In this study, we evaluated seven agronomic traits under well-watered (WW) and water-deficit (WD) conditions across two environments in a multi-parental DH population. In addition, two stress tolerance indices (STI and DRI) and the phenotypic plasticity were calculated in order to fully capture drought tolerance-related phenotypic diversity. In particular, the objectives of this study were to (1) assess phenotypic variation under two water conditions, (2) identify QTL and candidate genes associated with drought tolerance through GWAS using the three trait groups, (3) compare functional annotations of candidate genes via GO enrichment analysis to assess consistency and divergence in biological pathways linked to drought response, and (4) examine expression patterns of important candidate genes for the three phenotypic trait groups to further validate their drought tolerance mechanisms. In summary, this study provides a better understanding and novel insights into the genetic basis of drought tolerance, towards molecular breeding for improving drought tolerance in maize.

## 2. Results

### 2.1. Phenotypic Variation and Heritability of Agronomic Traits

Best linear unbiased estimates (BLUEs) were calculated for each trait in two environments with two water conditions (well-watered, WW, and water-deficit, WD). Seven agronomic traits, namely plant height (PH), ear height (EH), ear length (EL), ear diameter (ED), row number per ear (RNPE), kernel number per row (KNPR), and grain yield per ear (GYPE), exhibited large phenotypic variation in two environments (Table 1). The variation values of each trait under WD conditions were generally lower than those under WW conditions. For example, the variation for the GYPE ranged from 29.9 to 253.0 g in WW condition, while it ranged from 24.4 to 142.0 g in WD condition. For all traits, the genetic variance and the genotype-by-environment interaction variance were significant. For the ratio of the genotype-by-environment interaction variance ($\sigma_{ge}^{2}$) and the genetic variance ($\sigma_{g}^{2}$), both WW and WD of ED showed the highest values above 0.50, followed by GYPE (WD), which was the highest with 0.37. The genetic variance of all traits was highly significant (*p* < 0.01). Under WW conditions, the heritability ranged from 0.58 for ED to 0.88 for PH. Under WD conditions, the heritability ranged from 0.53 for GYPE to 0.80 for EH (Table 1). All traits showed approximately a normal distribution under both water treatments (Figure 1a–g) and therefore do not need a transformation. In summary, there were significant differences in the seven agronomic traits between WW and WD conditions (*p* < 0.01), and all traits showed significantly higher values under WW than that under WD conditions (Figure 1a–g). This illustrates the reliability and high quality of the phenotypic data, which are consequently well suited for GWAS mapping.

### 2.2. Correlation Analysis Among Agronomic Traits, Stress Tolerance Indices, and Phenotypic Plasticity

The correlation analysis showed that the seven agronomic traits under WD conditions were significantly positively correlated with their respective phenotype under WW conditions (*p* < 0.001) across two environments, and the correlation coefficients were all greater than 0.6, with the highest correlation coefficient of 0.88 between EH (WD) and EH (WW), followed by PH (WD) and PH (WW) with 0.84. There was a significant positive correlation between PH and EH under WD and under WW conditions (*p* < 0.001). Among the yield component traits, ED (WD) had the least significant correlation with other traits (*p* < 0.001), it was only significantly positive correlated with RNPE and GYPE under both water treatments, as well as with ED (WW). RNPE (WW) had the most significant negative correlations with EH and EL under both water treatments, as well as with PH (WW) (*p* < 0.05). Notably, GYPE (WW) was significantly positively correlated with all other traits under both conditions (*p* < 0.05), GYPE (WD) also was significantly positively correlated with other traits except EH (WW) (*p* < 0.05) (Figure 2). Regarding the single environments, at Urumqi GYPE showed the most significant positive correlations with other traits. GYPE (WD) was positively correlated with all traits except PH (WW) and EH (WW) (*p* < 0.05). GYPE (WW) exhibited significant positive correlations with all traits except RNPE (WW, WD) (*p* < 0.001). RNPE (WW) had the fewest significant positive correlations with other traits (*p* < 0.05) (Appendix A). At the Yulin location, GYPE (WW) was significantly positively correlated with all other traits, followed by GYPE (WD) (*p* < 0.05), while RNPW (WW) showed the most negative correlations (*p* < 0.05) (Appendix A). Both single- and multi-environment analyses confirmed that the trait GYPE is a representative trait for drought tolerance.

Regarding the correlation between stress tolerance indices (DRI and STI) and linear plasticity (LP), PH (LP) and EH (LP) were highly significantly positively correlated (*p* < 0.001), while the linear plasticity of EL, ED, RNPE, KNPR and GYPE was significantly positively correlated with each other (*p* < 0.05). In addition, with the exception of KNPR, the linear plasticity of all other traits was significantly positively correlated with their corresponding STI indices (*p* < 0.01). Both EL (LP) and KNPR (LP) were significantly negatively correlated with DRI of EL, ED, RNPE, KNPR and GYPE (*p* < 0.05), while GYPE (DRI) was not correlated with any of them. PH (DRI, STI) and EH (DRI, STI) were all significantly positively correlated with each other (*p* < 0.001). GYPE (STI) was significantly positively correlated with STI and DRI of all traits (*p* < 0.01) (Appendix A), indicating that GYPE (STI) was representative for the stress tolerance indices.

### 2.3. Phenotypic Plasticity Estimation Through Finlay–Wilkinson Regression

Finlay–Wilkinson Regression can be used to measure linear plasticity of phenotypic plasticity by quantifying the individuals’ response to the environments. For ED, the largest environmental effect was observed for the environment 19YL.WW with around 0.5, whereas the smallest environmental effect was found for 19WLMQ.WD with a value of around −0.1. The lines “LA511” and “LA490” showed a stronger response to the environment, with a linear plasticity value of 1.40 and 1.36, respectively, which means that the ED increases by 1.40 and 1.36 cm per unit increase in environmental effect. The lines “LA118” and “LA133”, by contrast, were the least responsive to the environment, with coefficients of 0.67 and 0.65, respectively (Figure 3a). For GYPE, the environmental effect of 19YL.WW was the largest with 25 and the environmental effect of 19WLMQ.WD was the smallest with about −20. The lines “C72” and “LA361” showed a stronger response to the environment, with a linear plasticity value of 3.31 and 3.20, respectively, illustrating that the GYPE increases by 3.31 and 3.20 for each unit increase in environmental effect, while the lines “LA473” and “LA49” had the lowest environmental response of 0.26 and 0.22, respectively (Figure 3b). This showed that the environment has a great influence on the GYPE, indicating that the yield is susceptible to the environment. For the remaining five traits, there was a large difference in phenotypic plasticity (Appendix A), which reflects the complexity of plant responses to environment. Therefore, further understanding of the genetic architecture underlying this is needed.

### 2.4. Population Structure and Linkage Disequilibrium of the Multi-Parental DH Population

A total of 167,809 high-quality single nucleotide polymorphisms (SNPs) were assembled and referenced to the B73_RefGen_v4 reference genome, after removing markers with a minor allele frequency (MAF) < 0.05 or a missing rate > 20.0%. The highest number of SNPs (26,928) was on chromosome 1, whereas the lowest number (12,561) was on chromosome 10, but the SNP marker density was generally uniform across the whole genome (Appendix A). Cluster analysis revealed five groups (Figure 4a), which was consistent with the genealogical information of the groups (Appendix A). The first three principal components explained 9.4%, 7.2% and 6.7% of the variation, and the first five principal components jointly explained more than 30% of the variation. Using the first four principal components, the population can be clearly classified into five groups (Figure 4b). The result of the principal component analysis was consistent with the evolutionary tree and genealogical information. Linkage disequilibrium (LD) decayed below $r^{2}=0.2$ after a physical distance of about 250 Kb (Figure 4c), indicating the potential for high-resolution GWAS mapping.

### 2.5. Genome-Wide Association Study (GWAS) for Three Trait Groups

The result of the GWAS revealed a total of 372 QTL for the three groups of phenotypes (seven agronomic traits under two water conditions, stress tolerance indices and phenotypic plasticity), which were identified on all ten chromosomes (Figure 5a). In total, across traits, 69 significant associations were identified under WW conditions, 61 under WD conditions, 101 for STI, 70 for DRI and 71 for phenotypic plasticity (Appendix A). The seven agronomic traits are mainly divided into two major categories, the phenological traits (PH, EH) and the yield components traits (KNPR, RNPE, EL, ED, GYPE).

#### 2.5.1. GWAS for Seven Agronomic Traits Under Water-Deficit (WD) and Well-Watered (WW) Conditions

A total of 130 QTL were identified under two water conditions. Of the 61 QTL identified under WD conditions, 13 were identified for the phenological traits of PH and EH, the remaining were associated with the yield component traits. Among these QTL, EL had most QTL with 21, while 6 QTL were found for ED, 7 for GYPE, 10 for KNPR, and 4 for RNPE (Appendix A). In addition, 69 QTL were detected under WW conditions, 11 for the phenological traits, the other 58 for the yield component traits. Interestingly, the GYPE trait had the highest number of QTL, totaling 33, accounting for almost half of the QTL found under WW conditions. In total, 2 QTL were associated with ED, 7 with EL, 9 with KNPR, and 7 with RNPE (Appendix A). Under WD conditions, the highest proportion of phenotypic variation explained (PVE) was found for qWD20 (Chr10: 76.54–76.55 Mb) associated with EL under WD, with 14.2%, and the $-{log}_{10}(p)$ of this QTL was the highest with 14.47. In total, there were 5 QTL with PVE greater than 10%, located on chromosomes 2, 4, 7 and 10. For the WW condition, the PVE was the highest for qWW34 on chromosome 5 associated with GYPE (WW), with 17.6%. In total, three QTL located on chromosomes 3, 4, and 5, had more than 10% PVE under WW conditions (Appendix A).

#### 2.5.2. GWAS for Stress Tolerance Indices

A total of 171 QTL were identified for the stress tolerance indices, 101 of them for STI and 70 for DRI. Among the STI QTL, 17 were for EH and 3 QTL for PH, and 81 QTL were identified for the yield component traits. For STI, EL had the largest number of QTL with 30, 11 QTL were detected for STI of ED, 17 for GYPE, 12 for KNPR, and 11 for RNPE (Appendix A). The PVE of qSTI14 associated with STI of EH on chromosome 2 was highest with 17.7%. A total of four QTL have a PVE greater than 10% (Appendix A). For DRI, there were 9 QTL for EH and 9 for PH, while for the yield component traits there were 52 QTL. Most of the DRI QTL were again found for EL with 27, the remaining were 3 QTL for DRI of ED, 12 for GYPE, 5 for KNPR, and 5 for RNPE (Appendix A). The highest PVE was found for qDRI58 on chromosome 3 associated with DRI of PH, explaining 14.8%. There were five QTL with PVE greater than 10% (Appendix A).

#### 2.5.3. GWAS for Phenotypic Plasticity

For the phenotypic plasticity, 71 QTL were identified. Three of them were found for EH and two for PH, the remaining 66 QTL were detected for the yield component traits (Appendix A). GYPE had the largest number of QTL with 33, three were associated with ED, 5 with EL, 18 with KNPR, and 7 with RNPE. The largest PVE (17.3%) was found for qLP4 (chr1: 271.96–272.15 Mb) associated with EH. A total of nine QTL explained more than 10% of the phenotypic variance, and these QTL were distributed on chromosomes 1, 2, 5, 9, and 10, respectively (Appendix A).

#### 2.5.4. QTL Identified for Agronomic Traits, Stress Tolerance Indices and Phenotypic Plasticity

Comparing the QTL results for the agronomic traits, the stress tolerance indices and the phenotypic plasticity, the number of QTL was smallest for the phenotypic plasticity. For example, there was only one QTL on chromosomes 3, 6 and 7 for plasticity, but more for the other trait groups (Figure 5a). The mapping for the three groups of traits showed that, except for GYPE and KNPR which had similar QTL numbers for the three trait groups, the stress indices had the highest numbers QTL for all other traits, followed by the agronomic traits (Figure 5b). These results indicated that more novel QTL could be identified by using stress indices than by directly measuring traits, while the QTL for the phenotypic plasticity were more difficult to detect.

Regarding all identified QTL, we found that only one QTL was co-localized among the three trait groups, one was found for phenotypic plasticity and stress indices, three were in common between phenotypic plasticity and agronomic traits, and 25 overlapped between stress indices and agronomic traits (Figure 5c, Appendix A). The results indicated that the genetic mechanisms of the three groups of traits are different, but that stress indices and agronomic traits may share some mechanisms.

### 2.6. Candidate Genes Functional Annotation

A total of 533 candidate genes were obtained for three groups of traits (Appendix A), among them, 86 for the WD condition, 91 for the WW condition, 112 and 134 for DRI and STI, respectively, and 110 candidate genes were identified for the phenotypic plasticity QTL. We performed Gene Ontology (GO) enrichment analysis of the candidate genes for the three trait groups and compared the differences in significantly enriched items. For the GO analysis, with *p* < 0.05 as the screening criterion, the top 20 enriched items were selected according to the significance degree to draw the bubble chart (Figure 6a–d).

The GO analysis of candidate genes of the seven agronomic traits under WD conditions (*p* < 0.05, Figure 6a) showed firstly that the enriched items were mainly in the aspect of environmental stress responses (e.g., reactive oxygen species, salt stress, inorganic substance, osmotic stress, hydrogen peroxide). Secondly, it was enriched in epigenetic regulation items like histone demethylation. In addition, some genes were enriched in protein dynamics and the formation of complexes. Others were enriched in metabolism (lignin metabolic) and enzyme activity (ent-copalyl diphosphate synthase activity). In conclusion, these results indicated that cells regulate and maintain stability in response to environmental stress under WD conditions through stress responses, epigenetic modifications, protein homeostasis regulation, and metabolic processes.

The GO analysis of candidate genes for the seven agronomic traits under WW conditions (*p* < 0.05, Figure 6b) showed that most of the significantly enriched items can be summarized as protein localization and transport. Secondly, a part of the significant enrichment was in RNA modification and enzyme activity regulation. In addition, some were enriched with cell development and differentiation, others were enriched in cellular defense and dynamic balance. Under WW conditions, these results reflect the synergistic mechanisms of cells in protein synthesis and localization, RNA metabolism, and developmental regulation.

For the stress tolerance indices, the GO analysis (*p* < 0.05, Figure 6c) revealed that most of the candidate genes were significantly enriched in the transportation of amino acids and organic acids. Secondly, some were mainly significantly enriched in histone modification (H3K4/H3K36 demethylation) and epigenetic regulation, similar to the findings for the QTL under WD conditions. In addition, some were mainly significantly enriched in terms of enzyme activity and RNA modification, similar to the enrichment for WW conditions. Other candidate genes were significantly enriched in organelle functions and oxidative stress responses. In short, these findings jointly revealed the synergistic mechanism of cells in metabolite transport, epigenetic regulation, enzyme catalysis and stress response.

The GO analysis of the candidate genes for the phenotypic plasticity QTL (*p* < 0.05, Figure 6d) showed that firstly, they are mainly significantly enriched in terms of environmental stress and adaptive responses (cold, high light intensity, UV, phosphate starvation), this may be due to a series of complex response mechanisms to other stresses caused by drought stress. Secondly, some were mainly significantly enriched in terms of metabolism (long-chain fatty acid catabolic and GDP metabolic) and enzyme activity (xylanase activity and guanylate kinase activity). In addition, some were significantly enriched in signal transduction and molecular interactions, some are significantly enriched in the regulation of cell growth and development, others were mainly reflected in the response and regulation network through fine feedback with positive or negative regulation. These results for linear plasticity reflect adaptive strategies to environmental fluctuations while maintaining developmental balance.

### 2.7. Mining of Candidate Genes and Identification of Polymorphisms

A total of 12 maize–rice drought tolerance homologous genes were obtained for the QTL for agronomic traits under two waters conditions, stress tolerance indices and phenotypic plasticity (Table 2). Among them, four genes mapped for linear plasticity, two co-localized with both stress tolerance indices and agronomic traits under WD, one co-localized with stress tolerance indices of different traits, and two were identified only by DRI. Additionally, two were associated solely with agronomic traits under WD, and one was related to under WW conditions.

In addition, we found that the 243.47–244.31 Mb region on chromosome 2 was co-located for both GYPE (WD, DRI, STI) and KNPR (WD, DRI, STI) traits (Appendix A), which co-located 8 QTL within 1 Mb and thus the largest number, making it an interesting target for further studies. The region of 244.25–244.31 Mb chromosome 2 was associated with GYPE (WD, DRI, STI) (Figure 7c), which overlapped the interval of 243.47–244.31 Mb for KNPR (WD, DRI, STI) (Figure 7a). Furthermore, the LD plot revealed a strong LD within this 0.84 Mb region. There were 24 candidate genes in this LD interval, among which one is a homologous gene of a rice drought tolerance gene. In this LD interval, there are five SNPs associated with KNPR and three associated with GYPE. Except for the SNP associated with STI of GYPE having no significant difference between the two alleles, the significant SNPs for KNPR (S2_243,465,767, S2_243,906,771, S2_244,306,161) under WD condition, DRI of KNPR (S2_243,475,944) and STI of KNPR (S2_244,302,606) showed significant differences (*p* < 0.05) between the two alleles (Figure 7b). The SNP S2_244,306,161 was associated with GYPE (WD, DRI) and had a significant effect (Figure 7d).

### 2.8. Analysis of Transcript Expression in Different Tissues and Under Drought Stress

The normalized log_2_(FPKM + 1) values from the public transcriptome datasets (SRP062027) were displayed on the heat map (Figure 8). The transcriptome expression of 35 important candidate genes, including genes of promising LD intervals (243.47–244.31 Mb) on chromosome 2 and maize–rice homologous genes of three groups of phenotypes, were analyzed for their expression patterns during the four stages (V12, V14, V16 and R1) of leaf, ear, and tassel tissues under drought stress, and the expression of 26 candidate genes was obtained by deleting the value of expression less than 2.

During drought stress, the results showed that there were five distributions: some genes were highly expressed independently in leaf, ear or tassel, others were highly expressed in the ear and tassel, and the remaining genes were expressed irregularly in various tissues (Figure 8). Among them, *Zm00001d028574* was most highly upregulated at various stages of leaf and ear, it was 9.0 times as high under drought stress than under the control in V12 leaf tissues. In addition, *Zm00001d052738* showed a strong expression in response to drought stress in R1 leaf and ear tissues, its expression level under drought condition was 4.0 times higher than under normal conditions in R1 leaf, and the expression level of drought stress in R1 ear tissues was 5.7 times that in normal conditions. While *Zm00001d051804*, *Zm00001d007954*, *Zm00001d007962*, and *Zm00001d016105* had been shown as a downregulated gene in response to drought at all stages of leaf tissue, *Zm00001d026501* was significantly upregulated in leaf tissue, but downregulated in tassel and ear. The expression level of *Zm00001d046501* was relatively high in the R1 stage of ear tissue and was obviously downregulated when subjected to drought stress (Appendix A).

## 3. Discussion

### 3.1. Combining Agronomic Traits, Stress Tolerance Indices and Phenotypic Plasticity Provides a New Vision for Drought Tolerance-Related Phenotypic Diversity

Drought tolerance is a complex quantitative trait that is regulated by numerous genes with minor effects. Consequently, a comprehensive evaluation method is better suited to assess this trait than a single phenotypic evaluation. In this study, we therefore combined agronomic traits, stress tolerance indices, and phenotypic plasticity to conduct a comparative analysis of QTL mapping in order to better understanding the genetic mechanism and underlying genes of drought tolerance. First, mapping of agronomic traits aimed to analyze the genetic control of agronomic traits under WD conditions. Previous studies had made significant progress in identifying candidate genes associated with various agronomic traits under WD conditions [62,63] and understanding the genetic basis behind their regulation. In addition, drought stress significantly impacts various agronomic or physiological traits of maize, including plant height [64], ear height [65], anthesis to silking interval [11], kernel row number [66], root traits [67], yield and its components [68], with the former traits ultimately also affecting grain yield [69]. In this study, we selected these traits, including the phenological traits PH and EH, and yield components traits (KNPR, RNPE, EL, ED, GYPE) for GWAS. The seven agronomic traits all showed responses to drought stress. Thus, phenological and yield component traits can be used as selection criteria since they are key traits examined in drought tolerance studies [70,71]. In conclusion, these agronomic traits can serve as targets for indirect selection for drought tolerance and can accelerate the generation of maize lines with drought tolerance and excellent agronomic performance.

Stress tolerance indices like STI and DRI can quantify stress response, thereby transforming complex stress responses into measurable genetic signals through growth or physiological indices. Stress indices can be used to screen germplasm for high drought tolerance, and provide targets for molecular marker-assisted breeding. Previous studies were widely used different stress tolerance indices to map QTL of drought tolerance [72,73]. Nouraei et al. (2024), for example, evaluated the stress tolerance indices (STI) and identified candidate genes in wheat [24]. Zhang et al. (2023) identified 54 loci by the drought resistance index (DRI) combined with GWAS in maize [26]. In the present study, we chose the indices STI and DRI, which were calculated in each environment as the ratio of the two treatments. The stress indices and the agronomic traits under the two water conditions had 25 co-located QTL (Figure 5c). This indicates that there are many multi-effect QTL between stress indices and agronomic traits, and that they in part share genetic mechanisms. On the other hand, 146 QTL were unique for the stress indices, indicating that stress indices can serve to identify new drought tolerance-associated genomic regions. The combination of stress indices and agronomic traits can more accurately evaluate maize responses to drought stress, and breeders can accelerate the selection of superior agronomic traits combined with drought tolerance by using the co-localized genomic intervals.

The QTL for phenotypic plasticity is mainly used to explain the genotype and environment interaction (G×E) effect. The genotype-by-environment interaction plays an important role in crops [31]. Phenotypic plasticity is a G×E effect in a broader sense, which provides a basis for predicting phenotypic response under climate change, and breeding of varieties adapted to a wide range of environmental conditions. However, so far, our knowledge on the genetic mechanisms of G×E is very limited. Some studies of phenotypic plasticity using linear plasticity (LP) were reported in maize [43], tomato [74] and wheat [75]. In this study, the largest number of QTL for LP was identified for GYPE. A total of 71 QTL were obtained for the LP of GYPE, while there were only 33 QTL for GYPE itself. This indicates that GYPE has rich phenotypic plasticity variations that can be further analyzed for genetics mechanisms. In general, however, less QTL were identified for LP than for the agronomic traits or the stress indices, indicating that the mapping of G×E is more difficult. In addition, phenotypic plasticity co-located one QTL with stress indices and agronomic traits, and showed three overlapping QTL with the agronomic traits, but the majority (94%) of the QTL for phenotypic plasticity are unique to G×E mapping. This indicates that the genomic regions of G×E mapped by the phenotypic plasticity are not identical with the location of the main genotype effects, and consequently, the corresponding genetic mechanism are also different. Kusmec et al. (2017) results revealed that candidate genes associated with mean phenotypic and plasticity phenotypic are structurally and functionally distinct, suggesting independent genetic control [31]. Which is consistent with our research findings.

In conclusion, the mapping of agronomic traits and stress indices can serve marker-assisted breeding for abiotic stress tolerance, while the mapping of plasticity focuses on the prediction of the ability to adapt to dynamic environments. The three trait groups jointly construct the genetic mechanism of plant adaptation and drought tolerance, providing targets for precise breeding.

### 3.2. GO Enrichment Analysis of Candidate Genes Showed Significant Differences Between Agronomic Traits, Stress Tolerance Indices and Phenotypic Plasticity

In this study, GO enrichment analysis was performed for candidate genes for the agronomic traits under two water conditions, the stress tolerance indices and the phenotypic plasticity. The results indicated the functional uniqueness of their respective candidate genes. The candidate genes for agronomic traits under WD conditions mainly responded to environmental stresses. This part of stress response was also mainly similar to the phenotypic plasticity in terms of environmental stress and adaptive responses. However, the candidate genes of phenotypic plasticity mainly showed GO enrichment in the responses to high light intensity, ultraviolet rays, cold and phosphate starvation. This might indicate that sensitivity to other stress was induced during drought stress, resulting in the activation of different stress response mechanisms. There were also some objective environmental factors due to the differences in light intensity, temperature and soil fertility between the fields in Urumqi (WLMQ; 87°61′ N, 43.79′ E) and Yulin (YL; 109°45′ N, 38°16′ E). Mapping of phenotypic plasticity reflects the complex response mechanism of plant drought tolerance and adaptive responses. Lee et al. (2024) found that sufficient light intensity was required for the drought responses in sweet basil [76]. Previous studies had shown that supplementation of phosphate can effectively modulate drought-induced oxidative stress by influencing antioxidant defense metabolism, which significantly improved drought stress in rice seedlings [77]. Nagatoshi et al. (2023) found that mild drought reduces inorganic phosphate levels in the leaves to activate the phosphate starvation response in soybean plants in the field, and this plays an important role in plant growth under mild drought [78]. Sun et al. (2021) demonstrated that phosphate transporter *MdPHT1;7* enhanced phosphorus accumulation and improved low phosphorus and drought tolerance [79]. The results of phenotypic plasticity showed a unique response regulation, such as positive regulation of response to extracellular stimulus, negative regulation of unidimensional cell growth, or positive regulation of cellular response to phosphate starvation. This reflects its fine coordination of environmental signals through responses to multiple environments.

Secondly, the candidate genes of agronomic traits under WD conditions are mainly involved the demethylation of histones, this was similar for the stress tolerance indices, indicating that epigenetic regulation has a unique response to the direct measurement of phenotypes and stress indices of drought stress response. Epigenetic modulation has emerged as a major factor in the transcriptional regulation of drought stress-related genes [80]. Yadav et al. (2025), for example, found that DNA methylation at cytosine residues might regulate drought-responsive genes under drought stress in chickpea [81]. Wang et al. (2021) showed that the Jumonji domain-containing H3K9 demethylase JMJ27 positively regulates drought stress responses through its histone demethylase activity in Arabidopsis [82]. These studies indicate that DNA methylation and/or histone modifications play a critical role in plant tolerance to drought stress. The analysis of candidate gene for stress tolerance indices suggests that these are involved in the transport of amino acids and organic acids, involving the dynamic balance of membrane protein activity and metabolites, and may be related to the regulation of metabolic intermediate products and energy metabolism regulation. The plants will accumulate a large amount of amino acids under stress, and these amino acids mainly act as osmolyte, regulation of ion transport, and detoxification of heavy metals, and they also affect the synthesis and activity of some enzymes, gene expression, and redox-homeostasis [83]. Diniz et al. (2020) found that amino acid and carbohydrate metabolism were coordinated to maintain energetic balance during drought in sugarcane [84]. Finally, the characteristics of candidate genes identified for agronomic trait under WW conditions are mainly reflected in protein synthesis and localization, RNA metabolism, and cell development regulation related to growth. In summary, the GO enrichment analysis reflected the different genetic mechanisms of the three trait groups in response to water stress.

### 3.3. Drought Tolerance Candidate Genes Were Identified by GWAS at the Harvest Stage

Multi-parental populations are powerful tools for GWAS mapping studies [44,49]. For example, Diouf et al. (2020) effectively identified the candidate genes for flowering time by using the average phenotype and phenotypic plasticity in a multi-parental tomato population [74]. We identified 35 potential candidate genes for the QTL for phenotypic plasticity, stress tolerance indices, and agronomic traits, all of which serve to further explore the molecular mechanism of drought tolerance in maize. Our analysis of overlapping QTL and the functional annotation results indicate that stress tolerance indices and agronomic traits in part share genetic mechanisms, but most candidate genes of the three trait groups are unique. Based on these findings, we recommend breeders prioritize candidate genes from both shared QTL regions (between stress indices and agronomic traits) and plasticity-specific loci for more precise trait improvement.

For GWAS analysis, we selected appropriate analysis models and relatively relaxed significance thresholds, as well as established screening criteria for candidate genes to ensure the reliability of the results. BLINK employs a multi-locus mixed model approach that effectively controls for population structure and polygenic background, improving detection power for true associations while reducing false positives [85]. Furthermore, it can overcome limitations of single-locus models that favor large-effect loci [86]. The conventional significance threshold in GWAS is highly stringent to control false positives but often leads to false negatives by missing true associations with small to moderate effects. Due to the use of multi-parental DH population, the genetic distances are relatively close, and it is not a purely natural population, so we employed the suggestive significance threshold approach [87]. In the process of screening drought-tolerant candidate genes, firstly, rice and maize share ancestral genome structure in grass crops with strong conservation characteristics [88], and there are more studies on the drought-tolerant genes in rice, which allows to identify their maize homologs. Secondly, considering that the LD intervals with the largest number of co-localizations have a greater possibility of drought tolerance.

There were 11 maize–rice drought-tolerant homologous genes (Table 2), some genes as transcription factors, such as WRKY, MYB and HSF, play important roles in drought tolerance [53,55,57]. Among them, four candidate genes were only mapped by plasticity indices. For example, *Zm00001d023262* (*brk3*) to be involved in the formation of epidermal cells of leaves and the properly polarized divisions of stomatal subsidiary mother cells in maize [89]. Its homologous gene *DS8* has a similar function in rice, and affects drought sensitivity by its involvement in leaf epidermal development and stomatal closure [50]. This gene was obviously upregulated at the R1 stage of leaf tissue responding to drought stress in transcriptome data. *Zm00001d028574* encodes a protein phosphatase homolog that has protein phosphatase 2C (PP2C) phosphase activity. Its homologous gene in rice, *OsPP2C09*, is a negative regulator of ABA signaling, and plays an important role in drought resistance [51]. Under drought stress, *Zm00001d028574* increased significantly in leaf tissues and showed the highest level expressed in the transcriptome. This expression was consistent with previous studies [90]. In addition, two genes (*Zm00001d016105* and *Zm00001d002545*) were only located by DRI, while two (*Zm00001d051804* and *Zm00001d017294*) also were only located by agronomic traits under WD conditions. Among them, overexpression of *Zm00001d016105* (*ZmPY10*) significantly increased the susceptibility of transgenic plants to ABA [91] and may improve drought tolerance of maize. Its homologous gene *OsPYL10* showed greater sensitivity to ABA in transgenic lines, and enhanced tolerance to drought stress in rice [54]. In maize, this gene was found to be downregulated in leaf tissue in response to drought stress in the transcriptome. The two genes, *Zm00001d052738* and *Zm00001d026501*, were candidate genes for co-localization of stress indices and agronomic traits under WD conditions, which is consistent with the conclusion that stress indices and agronomic traits share some genetic mechanisms in this study. For example, *Zm00001d052738* (ZmHsf11) associated with EL (STI and WD) trait, Qin et al. (2022) found that *ZmHsf11* decreased plant tolerance to heat stress by negatively regulating the expression of oxidative stress-related genes, increasing ROS levels and decreasing proline content in maize [92]. Its homolog gene is *OsHsfB2b*, and *OsHsfB2b*-RNAi transgenic lines showed significantly increased tolerance to drought stress in rice [57]. In the maize transcriptome, it was upregulated 4-fold in the R1 phase of leaf tissue and 5.7-fold in the R1 phase of ear tissue, this was consistent in the ear with the highly expression responding to drought stress in previous studies [93]. This gene may therefore play an important role in the ear for drought tolerance at the reproductive stage of maize.

There were 24 candidate genes contained in the LD (243.47–244.31 Mb) region on chromosome 2, this interval contains co-localized QTL for both GYPE (WD, DRI, STI) and KNPR (WD, DRI, STI), including agronomic traits and stress indices. There were eight pleiotropic QTL within the 1 Mb (Appendix A), indicating that this interval has a high probability of influencing two trait groups. Through analysis, three key candidate genes were finally determined. First, *Zm00001d007962* is a G2-like-transcription factor, associated with GYPE (STI) and KNPR (STI). Previous studies reported that *Zm00001d007962* was identified as water and nitrogen responsive genes by co-expression network analysis [94]. Its homologous gene *OsHHO3* regulates stomatal aperture by coordinating red light and abscisic acid in rice, which plays an important role in drought tolerance [58]. In the maize transcriptome, this gene was downregulated in leaf tissue in response to drought stress. In addition, *Zm00001d007949* and *Zm00001d007951* are candidates for QTL associated with KNPR under WD conditions. The lead SNP (S2_243906771) is located in the intron of *Zm00001d007949* and close to *Zm00001d007951*. *Zm00001d007949* (*zap1*) is a MADS-box transcription factors with a broad range of differentiated functions involved in spikelet and floral organ development [95]. *Zm00001d007951* (*ZmCK2α-1*) encodes CK2 protein kinase alpha1 that interacts with *DEK58* and *ZmSSF1* and these three proteins are essential for seed development in maize [96]. In the transcriptome data, *Zm00001d007949* was the highest in the ear and not expressed in the leaf. *Zm00001d007951* was all high in the leaf, ear, and tassel tissues, but it was more prominent in the ear, and the response to drought stress was slightly upregulated in all three tissues. *Zm00001d007949* and *Zm00001d007951* were extremely related to ear development, which suggests that both might affect the development of KNPR of the ear when subjected to drought stress.

Last, as for most QTL mapping studies, this study inherently has some limitations. A larger population size could increase the QTL detection power and the mapping resolution, and more environments might benefit G×E mapping. Nevertheless, in our study, the heritability across environments was moderate to high and thus well suited for a preliminary exploratory analysis as presented here. In the future, we plan to conduct more refined QTL localization analyses on the extreme materials selected in this study in more environments. In addition, the candidate genes presented here may include false positives and consequently, their functions need to be further verified in the future.

## 4. Materials and Methods

### 4.1. Plant Materials and Field Experiments

A multi-parental double haploid (DH) population of 196 lines derived from more than 30 breeding units (Appendix A), which were elite lines with extensive genetic diversity from a national joint breeding project (2016–2020) in China. They were broad and representative to be used for GWAS in this study. The multi-parental DH population was grown in the 2019 season at different experimental stations in China, namely at Urumqi (WLMQ; 87°61′ N, 43°79′ E) in Xinjiang and Yulin (YL; 109°45′ N, 38°16′ E) in Shaanxi, the two environments were designated 19WLMQ, 19YL. Urumqi and Yulin are key maize-producing regions in Northwest China, where drought-tolerant hybrids are urgently needed. Poor water-holding capacity and low fertility due to coarse-textured subsoil layers (sandy loam to loamy sand) and classified as Aridisols in Urumqi, and the weather is a hyper-arid climate, with mean low annual precipitation (~260 mm) and high evaporation (~2200 mm/year) (https://www.cma.gov.cn/en/, accessed on 15 June 2025). In addition, soil type classified as Loessial soil in Yulin, long-term over-cultivation has led to soil degradation and reduced water-use efficiency. The weather is erratic annual precipitation (~400 mm) (http://sn.cma.gov.cn/, accessed on 15 June 2025), with frequent mid-summer droughts during maize flowering (July–August). The experimental design adopted an augmented alpha-design with two replications within each environment. Each block consisting 20 plots with 18 lines and 2 control materials (Zheng58 and Chang72), and each plot including two rows with 4 m in length, and 0.55 m row spacing. Plant spacing was 0.20 m, achieving a planting density of around 90,000 plants per hectare.

We have set up two types of treatments: the water area (well-watered, WW) and the dry area (water-deficit, WD). The two treatments areas were placed 3 m wide isolation belts to prevent water seepage, and a 5 m buffer zone was set up around the two treatments area to minimize edge effects. Separate water meters were installed for each treatment to precisely control irrigation volume. WW: Normal irrigation during the entire growth period, with a total of 10 drip irrigations and an average of 600 m^3^/ha per irrigation. The WD treatment period was imposed from late vegetative stage prior to tasseling until the end of the flowering stage. During WD, the irrigation volume was 300 m^3^/ha, while the other periods were the same as the normal treatment. Two treatments adopted drip irrigation under plastic mulch with integrated water and fertilizer management, with fertilizer applications split according to growth stages: basal compound fertilizer at planting, followed by urea at other stages.

### 4.2. Phenotypic Evaluation

During the harvest period, eight plants with normal growth conditions were randomly selected from each plot to assess phenological traits and calculate average values for plant height (PH) and ear height (EH), which were measured using a calibrated extendable ruler on the same day. The yield component traits, ear length (EL), ear diameter (ED), row number per ear (RNPE), kernel number per row (KNPR), and grain yield per ear (GYPE), were measured according to the number of plants with normal ear development in each plot. GYPE was adjusted to a 14% moisture content and was the mean value of the number of plants in a plot. To ensure data reliability, any ears with signs of disease, pest damage, or abnormal development were excluded from measurement. The brief methods and description of seven agronomic traits for the measure are listed in Table 3.

The stress tolerance indices were calculated as previously described, namely stress tolerance index (STI) [25] and drought resistant index (DRI) [26]. They are defined as ratios between the phenotypic BLUE value of a trait for a genotype under WD to WW condition. These stress tolerance indices were computed for each measured trait in each environment as follows:$$STI=\left(Yd\times Yw\right)/{Ymw}^{2}$$$$DRI=\left(Yd/Yw\right)\times \left(Yd/Ymd\right)$$
where *Yd* is the measured trait of a genotype tested under WD, *Yw* is measured trait of the same genotype tested under WW conditions, *Ymd* is the average measured trait of all genotypes under WS conditions, and *Ymw* is the average measured trait of all genotypes tested under WW conditions.

### 4.3. Statistical Analysis

First, the method of studentized residual razor was adopted to remove outliers in the original data [97]. We calculated the best linear unbiased estimates (BLUEs) for each measured trait by fitting a linear mixed model using the R package lmer [98]. The following model was used under WW and WD conditions in each single environment: *y* = *μ* + *G* + *Rep* + *Block*(*Rep*) + *ε*, where *y* represents the phenotype observation value, *μ* is the overall mean, *G* the genotypic effect, *Rep* the effect of the replication, *Block*(*Rep*) the block effect nested within the replication, and *ε* is the random error. The following model was used under WW and WD conditions for the multi-environment analysis: *y* = *μ* + *G* + *E* + *G×E* + *Rep*(*E*) + *Block*(*Rep*(*E*)) + *ε*, where *y*, *μ*, *G* are the same as in the above model, *E* is the environmental effect, *G×E* the interaction of the genotype-by-environment, *Rep*(*E*) the effects of replication in a particular environment; *Block*(*Rep*(*E*)) the block effect within a specific replication in a specific environment; *ε* is a random error.

The heritability of phenotypic data was estimated with the R package sommer [99] and the formula proposed by Cullis et al. (2006) [100] with the unbalanced data under multiple environmental conditions:$$H^{2}=1-\frac{{\bar{v}}_{BLUE}}{{2\sigma }_{G}^{2}}$$
where ${\bar{v}}_{BLUE}$ was the mean variance of genetic prediction error, and $\sigma_{G}^{2}$ was the genetic variance. Descriptive statistics of agronomic traits were calculated with R (version 4.3.1). Considering that the phenotype is linearly additive, when using QTL mapping, normality display is required. The frequency distributions histogram of the agronomic traits across two environments were visualized using R package ggplot2 (version 3.4.2) [101]. Pearson’s correlation coefficients were drawn using the R package PerformanceAnalytics (version 2.0.8).

### 4.4. Finlay–Wilkinson Regression Analysis

To estimate phenotypic plasticity, the sensitivities of an individual to the environment can be parameterized by Finlay–Wilkinson Regression [30]. The G × E effect was decomposed by plasticity using Finlay–Wilkinson Regression as:$$y_{ij}=\mu +g_{i}+(1+b_{i})h_{j}+\varepsilon_{ij}$$
where $y_{ij}$ is the BLUE value of the *i*th line in the *j*th environment, $g_{i}$ is the genetic effect of the ith line following the distribution *g*~*N* (0, *K*$\sigma_{g}^{2}$), K is the relationship matrix, $h_{j}$ is the effect of the *j*th environment following the distribution *h*~*N* (0, *H*$\sigma_{h}^{2}$), where *H* is environments of the covariance matrix, and $\varepsilon_{ij}$ the error following the distribution *ε*~*N* (0, *I*$\sigma^{2}$), the term (1 + $b_{i}$) can be interpreted as the linear plasticity of the *i*th line to the environment, (1 + $b_{i}$) represents the amount of phenotypic change per unit change in environmental effects, where *b*~*N* (0, *K*$\sigma_{b}^{2}$). The above model was solved in the R package FW [102] using a Bayesian method.

### 4.5. Genotyping

The 196 DH lines were genotyped based on maize 48K liquid phase probe capture technique. Heterozygous genotypes were considered as missing data, then the Beagle (version 5.4) [103] software was used for filling. Using vcftools (version 0.1.16) [104] software for quality control, individuals SNP with missing rate > 20%, and minor allele frequency (MAF) < 5% were removed. Physical positions of the SNP markers were obtained from the reference assembly B73 RefGen_v4 (https://ftp.ensemblgenomes.ebi.ac.uk/pub/plants/release-49/gff3/zea_mays/Zea_mays.B73_RefGen_v4.49.gff3.gz, accessed on 9 February 2025). Finally, 167,809 high-quality SNPs were used for subsequent analyses.

### 4.6. Population Analysis and Linkage Disequilibrium (LD) Analysis

The high-quality SNPs were used for the phylogenetic tree, principal component analysis (PCA), and linkage disequilibrium (LD) analysis. The genetic distances were calculated by the function “Distance Matrix” of Tassel (version 5.2.90) [105], then we used the function Neighbor Joining of “Create Tree” methods to construct the polygenetic tree, and visualization was performed by the iTOL site (https://itol.embl.de/, accessed on 9 February 2025). The PCA was calculated using PLINK (version1.9) [106] software, and the component number was set to five. R package ggplot (version 3.4.2) [101] was used to make the scatter plot. The LD decay with distance was calculated in PopLDdecay (version 3.40) [107], when $r^{2}$ is equal to 0.2, the corresponding distance is the LD.

### 4.7. Genome-Wide Association Study

Genome-wide association study (GWAS) was conducted by combining phenotypic data (across environments BLUE, stress tolerance indices, phenotypic plasticity) and genotype data. For GWAS, the BLINK model with PC was used in the R package GAPIT (version 3.0) [108]. The R package CMplot (version 3.6.2) [109] was used to plot the Manhattan and QQ plots of the results of the association analysis. It was too strict to calculate the Bonferroni-corrected threshold using all markers, as many were in strong LD with others, making them redundant markers. The independent marker number was instead used and calculated in the indep-pairwise module of PLINK (version 1.9) [106]. Finally, 10,910 independent markers were obtained. The suggestive threshold to control the type I error rate was global α = 0.10, then the significant threshold was $-{log}_{10}(0.1/\left(10,910/(10\times 2)\right))=3.74$ with chromosome-wide Bonferroni correction. With α=0.05, the significance threshold was 4.04.

### 4.8. Candidate Gene Identification and Functional Analysis

The confidence intervals of QTL were obtained by the following steps: (1) the significant SNPs were selected, (2) the distance between the physical location of the most significant SNP and other SNPs was calculated. If the distance was less than LD $r^{2}$ = 0.2 (250 Kb), the smaller SNP was deleted, and the remaining SNP was called the lead SNP. (3) Taking the lead SNP as the center, the boundary of LD $r^{2}$ = 0.2 was calculated from left to right, and the confidence intervals obtained by subtraction of the boundary distance was called the QTL. Any two QTL for which the confidence intervals overlapped or the peak position was within 1 Mb were regarded as pleiotropic QTL [32]. Then, to further identify the relationships between genes and drought-related traits, gene ontology (GO) enrichment analysis was performed using the online platform OmicShare (https://www.omicshare.com/tools, accessed on 9 February 2025) to investigate the functions of these genes, and the result was drawn using the R package ggplot (version 3.4.2) [101].

The potential candidate genes of agronomic traits, stress tolerance indices and phenotypic plasticity were selected following two standards: (1) Genomic regions with most pleiotropic QTL regions were considered as important candidate intervals as well as potential candidate genes. (2) The maize–rice homology genes, maize genes orthologous to rice drought tolerance genes, were identified by BLASTP sequence alignment. For this, we downloaded rice drought tolerance genes with functional verification from the China Rice Data Center (https://www.ricedata.cn/, accessed on 1 December 2024), and extracted the protein sequences of these genes through the Rice Genome Annotation Project (http://rice.uga.edu/analyses_search_blast.shtml, accessed on 1 December 2024). The BLASTP alignment of the maize–rice protein sequence was performed through the Phytozome website (https://phytozome-next.jgi.doe.gov/, accessed on 1 December 2024). The alignment criterion was Evalue < $1\times e^{-10}$ and identity > 40%.

### 4.9. Expression Analysis of Potential Candidate Genes

We conducted transcriptional analysis of the potential candidate genes in response to drought stress and downloaded the public transcriptome datasets of the accession number SRP062027 (NCBI Bio-project PRJNA291919: GEO accession GSE71723) from *Zea mays* RNA-seq database (https://plantrnadb.com/zmrna/, accessed on 1 December 2024) [110]. The PRJNA291919 RNA-seq data was used to analyze the expression pattern in well-watered and drought-stressed conditions and includes three tissues (leaf, ear, and tassel) at four developmental stages (V12, V14, V18, and R1) [111]. The genes with the transcripts per million (FPKM) values of ≥2 were considered to be expressed under water-deficit and well-watered condition, and the FPKM values were used to construct a heatmap using the R package pheatmap (version 1.0.12).

## 5. Conclusions

In this study, we performed GWAS analysis for drought tolerance in a multi-parental population of maize. This identified 130 QTL for agronomic traits under two water conditions, 71 for phenotypic plasticity, and 171 for stress tolerance indices. Among them, there was only one overlapping between all three trait groups, while most of QTL were unique for their group, indicating that the genetic mechanisms underlying the three trait groups are different. GO enrichment analysis of candidate genes also suggested different genetic mechanisms of these three groups of phenotypes in response to drought stress in maize. Collectively, our results provide further knowledge of the genetic control underlying drought tolerance in maize and can assist future marker-based selection of drought-tolerant varieties.

## Figures and Tables

**Figure 1 ijms-26-06285-f001:**
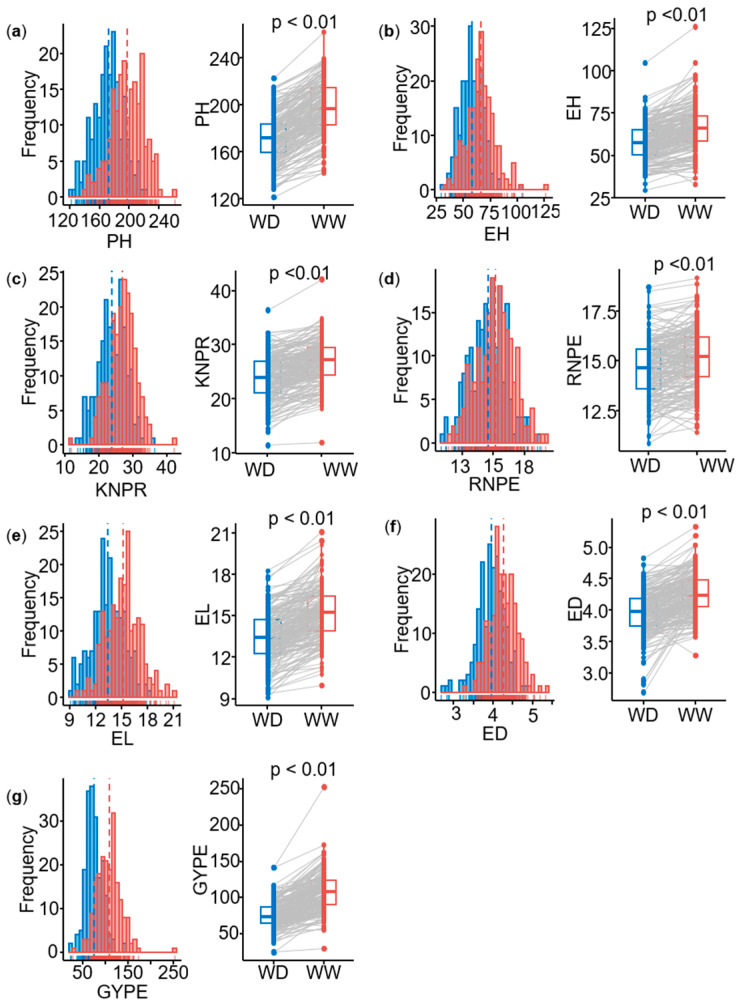
Distribution of seven agronomic traits of the multi-parental DH population under two water treatments. (**a**) PH: plant height (cm), (**b**) EH: ear height (cm), (**c**) KNPR: kernel number per row (count), (**d**) RNPE: row number per ear (count), (**e**) EL: ear length (cm), (**f**) ED: ear diameter (cm), (**g**) GYPE: grain yield per ear (**g**). Note: the significance of the difference was calculated by *t*-tests between the phenotypes under water-deficit (WD) and well-watered (WW) conditions for all traits. The blue distribution represents the phenotypic BLUE value under WD conditions, and the red distribution represents the phenotypic BLUE value under WW conditions.

**Figure 2 ijms-26-06285-f002:**
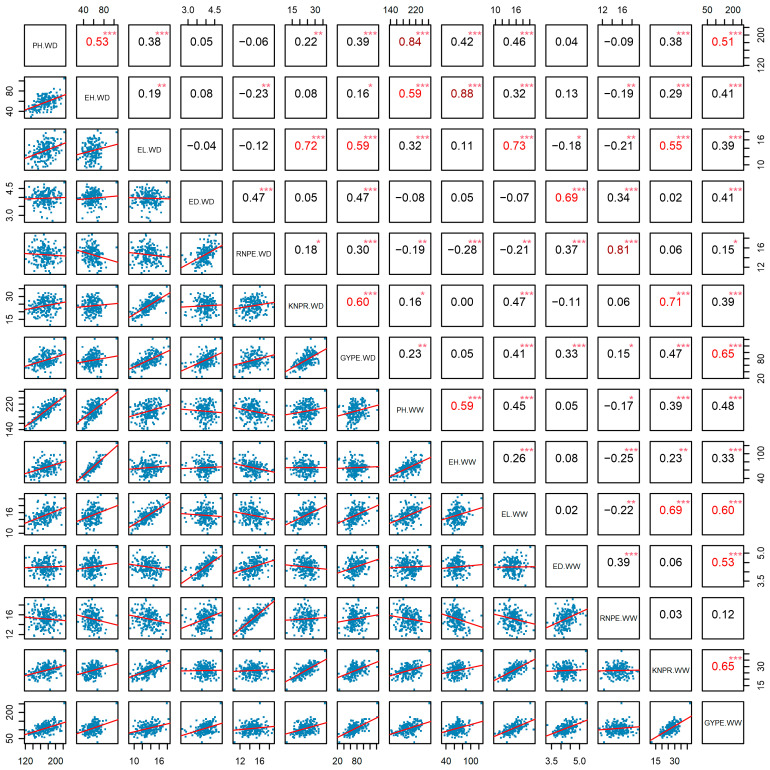
Correlations based on the phenotypic BLUE values of seven agronomic traits under well-watered (WW) and water-deficit (WD) conditions in the DH population. Note: *: *p* < 0.05; **: *p* < 0.01, ***: *p* < 0.001. In the upper right corner, the numbers of red or deep red indicate moderate or high correlation, respectively. The red lines represent the trend of correlation in the lower left corner. PH: plant height, EH: ear height, EL: ear length, ED: ear diameter, RNPE: row number per ear, KNPR: kernel number per row, GYPE: grain yield per ear.

**Figure 3 ijms-26-06285-f003:**
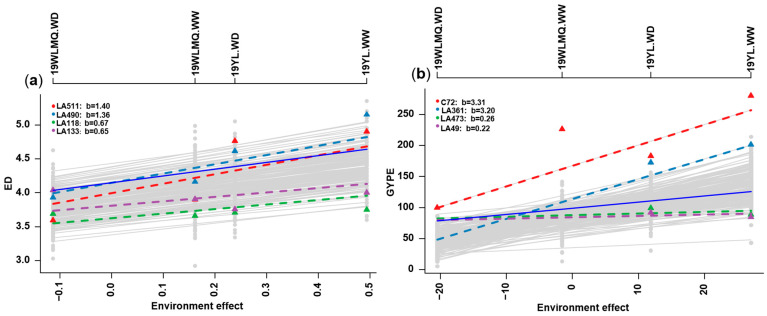
Environment effect and phenotypic plasticity. (**a**) The linear plasticity (LP) of ED, (**b**) LP of GYPE. Note: The linear plasticity values of the two highest and two lowest for the trait are shown. The dashed lines represent the slope (linear plasticity) of individual genotypes; the greater the slope, the greater the linear plasticity of the individual. The solid blue line represents a slope of one. ED: ear diameter, GYPE: grain yield per ear.

**Figure 4 ijms-26-06285-f004:**
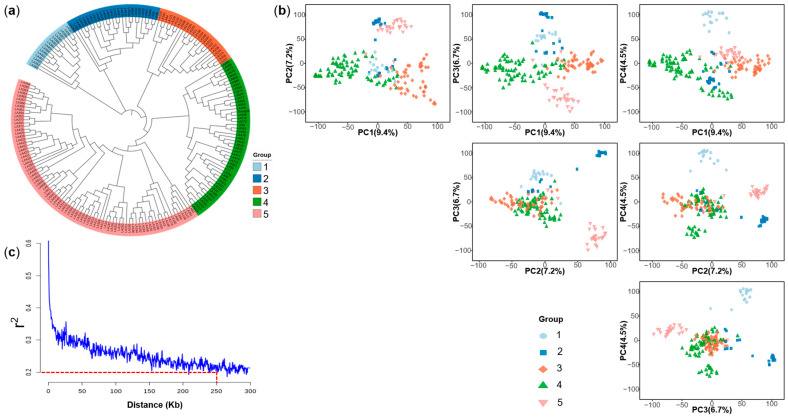
Population structure and linkage disequilibrium of the multi-parental DH population. (**a**) Neighbor-joining tree, (**b**) principal component analysis, and (**c**) decay of linkage disequilibrium with physical distance.

**Figure 5 ijms-26-06285-f005:**
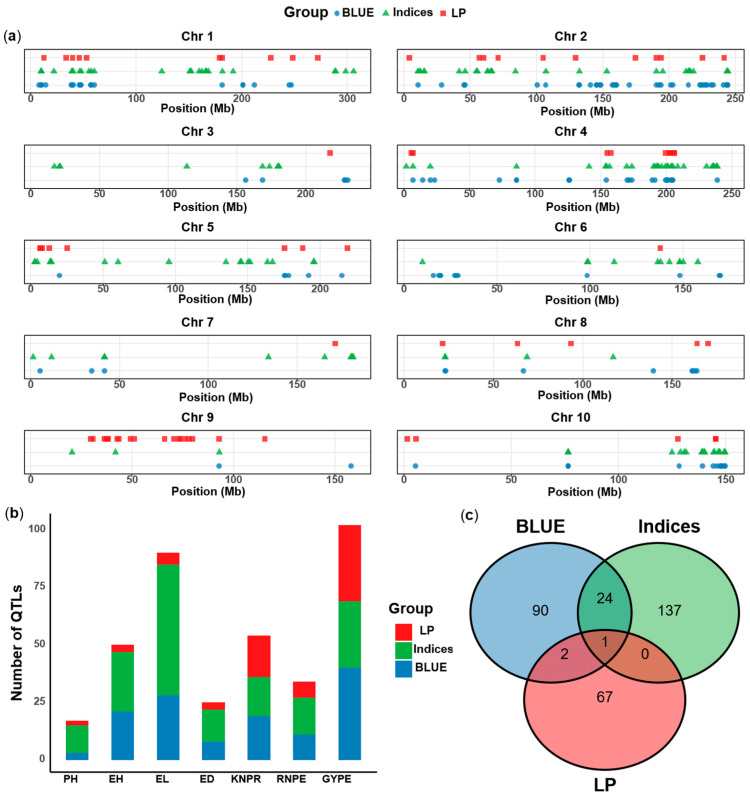
Comparison of physical maps on 10 chromosomes with QTL signals for the agronomic traits (BLUEs of two environments) under two waters conditions, stress tolerance indices and phenotypic plasticity. (**a**) Distribution of QTL on 10 chromosomes for the three groups of traits. (**b**) Distribution of QTL for the seven agronomic traits for the three group of phenotypes. (**c**) Venn diagram of the coincidence of the three trait groups. LP: phenotypic plasticity, Indices: stress tolerance indices, BLUE: agronomic traits under two waters conditions. PH: plant height, EH: ear height, EL: ear length, ED: ear diameter, KNPR: kernel number per row, RNPE: row number per ear, GYPE: grain yield per ear.

**Figure 6 ijms-26-06285-f006:**
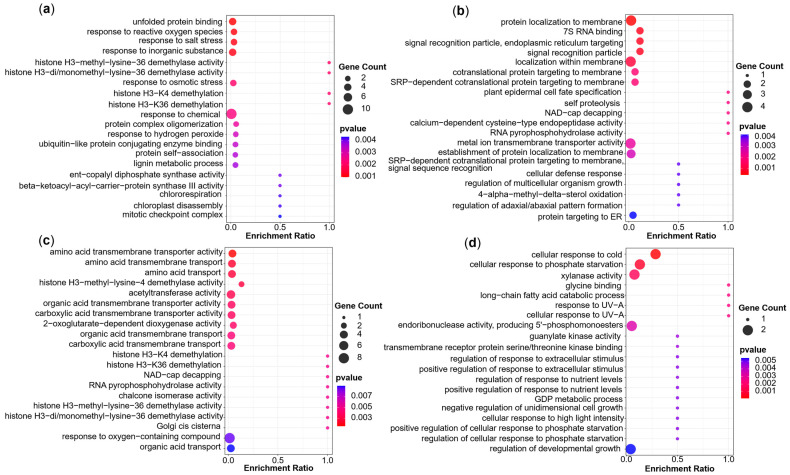
GO significant enrichment analysis of candidate genes (*p* < 0.05). GO analysis of candidate genes for (**a**) seven agronomic traits under WD conditions, (**b**) seven agronomic traits under WW conditions, (**c**) stress tolerance indices, and (**d**) phenotypic plasticity.

**Figure 7 ijms-26-06285-f007:**
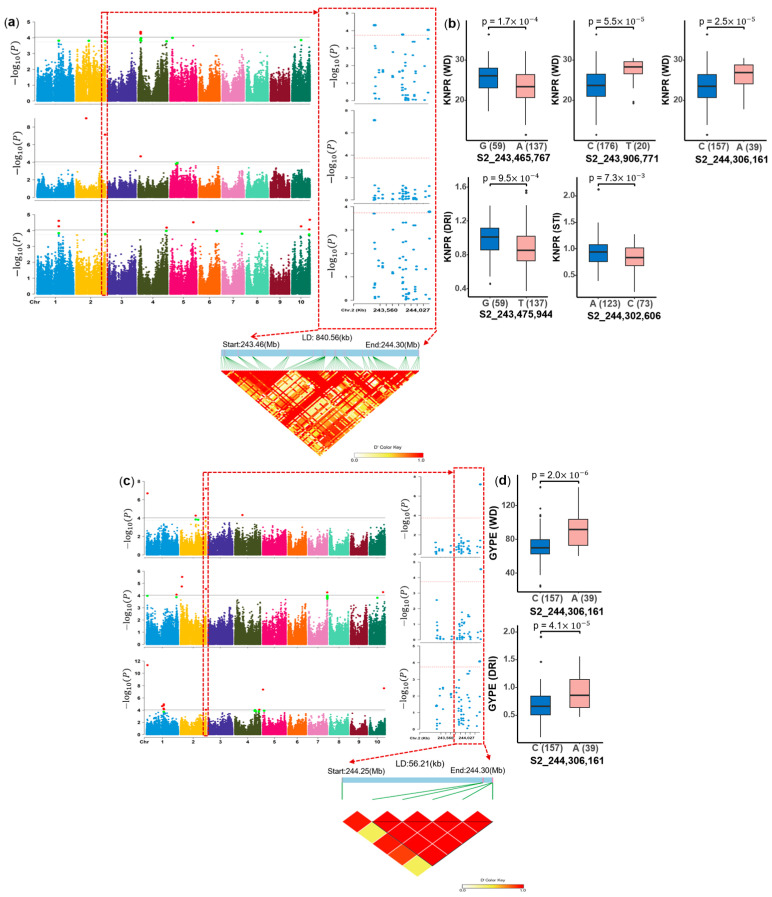
Genome segments and polymorphism analysis on chromosome 2 co-localization of highly associated hotpot QTL. (**a**) Zoom on the 243.47–244.31 Mb region associated with kernel number per row (KNPR), (**b**) polymorphism analysis by boxplot for 5 SNPs (S2_243,465,767, S2_243,906,771, S2_244,306,161, S2_243,475,944, S2_244,302,606) with significant effects (*p* < 0.05). (**c**) Locus zoom on 244.25–244.31 Mb region associated with grain yield per ear (GYPE). (**d**) Comparison of the allele effect for SNP S2_244,306,161 associated with GYPE (WD, DRI) (*p* < 0.05). Note: Manhattan plot (top) and LD heat map (bottom) of genome segments, the green dots represent $-{log}_{10}(p)$ values greater than the significant threshold of 3.74 and less than 4.04, while the red dots represent $-{log}_{10}(p)$ values greater than the significant threshold of 4.04. The ten chromosomes are represented by different colors respectively. the x-axis of boxplot represents the two alleles for each associated SNP, while the y-axis corresponds to BLUE values of the corresponding trait.

**Figure 8 ijms-26-06285-f008:**
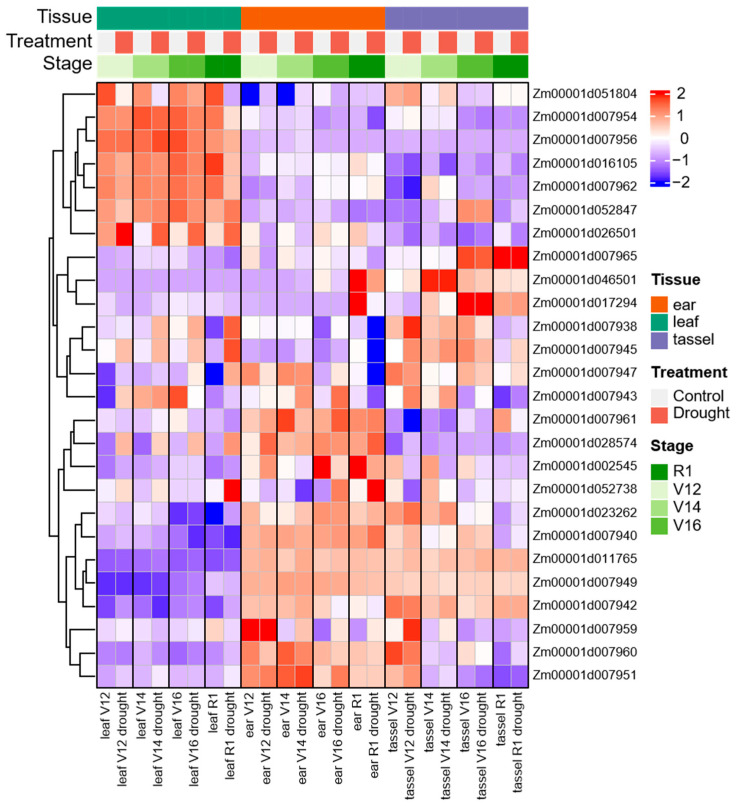
Relative transcript abundance of 26 candidate genes at the four developmental stages (V12, V14, V16, R1) of leaf, ear, and tassel tissues under control and drought stress conditions.

**Table 1 ijms-26-06285-t001:** Descriptive statistics of phenotypic data in two environments.

Trait	Treatment	Min	Max	Mean	SD	$\sigma_{g}^{2}$	$\sigma_{ge}^{2}$	Ratio	*H* ^2^
PH	WW	142.0	261.9	197.8	21.9	424.6 ***	15.4 **	0.04	0.88
PH	WD	121.7	222.7	172.0	20.1	277.1 ***	85.2 ***	0.31	0.71
EH	WW	32.9	126.0	66.2	13.4	146.9 ***	22.7 ***	0.15	0.81
EH	WD	29.5	104.6	57.6	10.7	90.4 ***	9.3 ***	0.10	0.80
EL	WW	10.0	21.1	15.2	1.9	3.0 ***	0.38 ***	0.13	0.82
EL	WD	9.1	18.3	13.4	1.8	2.2 ***	0.67 ***	0.30	0.69
ED	WW	3.3	5.3	4.3	0.33	0.07 ***	0.04 ***	0.57	0.58
ED	WD	2.7	4.8	4.0	0.35	0.08 ***	0.04 ***	0.50	0.64
RNPE	WW	11.4	19.2	15.2	1.5	1.6 ***	0.57 ***	0.35	0.72
RNPE	WD	10.9	18.7	14.6	1.5	1.5 ***	0.50 ***	0.33	0.69
KNPR	WW	11.9	42.1	26.9	4.0	12.8 ***	1.5 **	0.12	0.79
KNPR	WD	11.4	36.4	23.9	4.2	12.4 ***	0.78 *	0.06	0.72
GYPE	WW	29.9	253.0	108.5	26.4	555.9 ***	122.6 ***	0.22	0.76
GYPE	WD	24.4	142.0	75.7	18.4	179.1 ***	65.9 **	0.37	0.53

Note: Min: minimum value; Max: maximum value; SD: standard deviation; $\sigma_{g}^{2}$: genetic variance; $\sigma_{ge}^{2}$: the genotype-by-environment interaction variance; Ratio: the ratio between $\sigma_{ge}^{2}$ and $\sigma_{g}^{2}$; *H*^2^: broad-sense heritability of the trait. ***: 0 < *p* < 0.0001; **: 0.0001 < *p* < 0.01; *: 0.01 < *p* < 0.05. PH: plant height (cm), EH: ear height (cm), EL: ear length (cm), ED: ear diameter (cm), RNPE: row number per ear (count), KNPR: kernel number per row (count), GYPE: grain yield per ear (g). WW: well-watered, WD: water-deficit.

**Table 2 ijms-26-06285-t002:** Important candidate genes homologous to rice drought tolerance genes within QTLs.

Gene	Description	Class	Traits	QTL	Homology in Rice	Reference
*Zm00001d023262*	Brick3	LP	RNPE	qLP65	*OsDS8*	[50]
*Zm00001d028574*	Protein phosphatase homolog5	LP	EL	qLP7	*OsPP2C09*	[51]
*Zm00001d046501*	AP2-EREBP-transcription factor 237	LP	KNPR	qLP62	*OsWR1*	[52]
*Zm00001d052847*	WRKY-transcription factor 87	LP	GYPE	qLP26	*OsWRKY30*	[53]
*Zm00001d016105*	Pyrabactin resistance-like protein10	DRI	PH	qDRI59	*OsPYL10*	[54]
*Zm00001d002545*	MYB-related-transcription factor 20	DRI	EL	qDRI24	*OsMYBR1*	[55]
*Zm00001d026501*	Glutamine synthetase1	DRI	EL	qDRI19	*OsGS2*	[56]
		WD	EH	qWD13		
*Zm00001d052738*	HSF transcription factor 7	STI	EL	qSTI42	*OsHsfB2b*	[57]
		WD	EL	qWD31		
*Zm00001d007962*	G2-like-transcription factor 27	STI	GYPE	qSTI66	*OsHHO3*	[58]
		STI	KNPR	qSTI81		
*Zm00001d051804*	Glutamine synthetase5	WD	EL	qWD30	*OsGS1;1*	[59]
*Zm00001d017294*	Gibberellin 2-oxidase4	WD	EL	qWD35	*GA2ox6*	[60]
*Zm00001d011765*	Methylsterol monooxygenase 2-2	WW	GYPE	qWW46	*OsGL1-8*	[61]

Note: EH: ear height, EL: ear length, RNPE: row number per ear, KNPR: kernel number per row, GYPE: grain yield per ear. LP: Linear plasticity of phenotypic plasticity, DRI: drought resistance index, STI: stress tolerance index, WD: water-deficit, WW: well-watered.

**Table 3 ijms-26-06285-t003:** Description of the seven agronomic traits for drought tolerance in maize.

Class	Traits	Description	Measuring Unit
Phenological	PH	The height from ground to the tassel tip	centimeters (cm)
	EH	The height from ground to the node of the highest ear	centimeters (cm)
Yield	EL	The length of maize ear from base to top	centimeters (cm)
composition	ED	The diameter of the middle ear	centimeters (cm)
	RNPE	The number of rows of complete ear	count (rows/ear)
	KNPR	The kernel number per row	count (grains/row)
	GYPE	The yield per ear as the mean value of the effective plants in the plot	grams (g)

Note: For phenological traits, eight plants were randomly selected in the plot to measure the average value. For the yield component traits, the average value was calculated according to the number of effective plants.

## Data Availability

The datasets generated during and/or analyzed during the current study are available from the corresponding author upon reasonable request.

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
