# Peer review of "Genetic Dissection of Drought Tolerance in Maize Through GWAS of Agronomic Traits, Stress Tolerance Indices, and Phenotypic Plasticity"

_ijms, 2025, doi:10.3390/ijms26136285_

Round 1

Reviewer 1 Report

Comments and Suggestions for Authors

The manuscript has a scientific contribution but there are some queries that need to be addressed:

  • The abstract offers a clear summary of the research objectives and findings. However, some phrasing could be more concise.
  • Line 50: Please add more about critical stage sensitive to drought stress in maize.
  • In the introduction section, the authors should offer specifics about the knowledge gap that their study wants to fill.
  • Line 65: What links or new research did the researchers discover between this study and previous ones?
  • Line 153: I would recommend that correlation analysis be presented for each environment separately to better identify the key traits contributing to grain yield under drought stress. These clearer which traits are more influential under specific drought conditions.
  • Line 226: The high number of QTL for stress tolerance indices (171) compared to other categories (61–71) is striking. Is this due to trait complexity, or methodological bias (e.g., more traits in this category)?
  • Line 633: Please add the reason for choosing these 196 double haploids (DH) lines.
  • Line 635: give more details about the specific experimental station (e.g., soil type classification, previous cropping history) and reason for its selection are needed.
  • Line 643: This study is done in only one season (2019). Including one more year or season would improve the estimation of environmental effects, leading to more accurate trait values due to the influence of the environment genotype-by-environment interaction (GxE).
  • Line 639: What is the exact amount of applied water for WW and WD treatments?. How do the authors determine and validate drought stress conditions? Are soil water content or other measures used to confirm the level of stress?
  • In the Discussion section, it's better to avoid repetition of the results. Additionally, removing the subtitles could help streamline the narrative and enhance readability
  • Line 650: grain yield per ear or grain yield per plant.

Reviewer 2 Report

Comments and Suggestions for Authors

Dear Authors, 

Please find my observations for "Genetic Dissection of Drought Tolerance in Maize through GWAS of Agronomic Traits, Stress Tolerance Indices, and Phenotypic Plasticity" manuscript

  1. In my opinion the abstract should be shortened and become more targeted and concise
  2. The introduction provides background on drought tolerance in maize and the importance of GWAS, but it does not clearly articulate the specific research gap this study addresses.
  3. The introduction provides general background on maize, drought stress, and the importance of genetic studies, but it does not clearly state what is unknown or unresolved in the field. For example, while it mentions that “dissecting the genetic basis of drought tolerance is crucial,” (L57) it does not well specify what previous studies have failed to address or what unique aspect this study will tackle.
  4. Please improve in terms of clarity the objectives of the research (L98)
  5. In my opinion the "4.1. Plant Materials and Field Experiments" (L632) section lacks critical information about the field layout, how plots were randomized, how environmental heterogeneity was minimized, and whether blocking or spatial correction was used. 
  6. Please clarify if “two replicates,” (L642) refer to biological or technical replicates, and how they were arranged in the field. 
  7. Please provide more information referring to the water-deficit (WD) treatment as this in the current form of the manuscript is described as “half of the water volume of WW,” but the actual quantities, timing, and method of application are not clearly specified
  8. Please include more information about other environmental factors (soil type, fertilization, pest management), if they were controlled or monitored, as they could confound the results.
  9. The manuscript should better detail the calibration or quality control for phenotypic measurements
  10. Please mention how data normality, homoscedasticity was considered; or if it was a need for transformation 
  11. Please better describe the criteria for selecting candidate genes 
  12. The Results section presents a large volume of data in dense paragraphs, making it difficult for readers to quickly grasp the key findings. Please consider figures resolution also
  13. In my opinion the discussion sometimes presents the study’s results as broadly applicable or definitive, without well acknowledging the limitations of the experimental design, sample size, or environmental conditions.
  14. I recommend for authors to better consider the study’s limitations (functional validation for candidate genes, the potential for false positives in GWAS, limited number of environments and replicates, etc)
  15. Please better compare current manuscript findings with the most recent or relevant literature 
  16. The discussion mentions candidate genes and QTL, but in my opinion it does not provide enough in-depth biological interpretation or hypotheses about the mechanisms underlying the observed associations
  17. Please consider to include the rationale for certain statistical thresholds, GWAS models, or candidate gene selection criteria in the discussion section

Round 2

Reviewer 1 Report

Comments and Suggestions for Authors

The revised version is acceptable. 

Author Response

Thank you very much for taking the time to review this manuscript. 

Reviewer 2 Report

Comments and Suggestions for Authors

Dear Authors,

Thank you for considering my recommendations into "Genetic Dissection of Drought Tolerance in Maize through GWAS of Agronomic Traits, Stress Tolerance Indices, and Phenotypic Plasticity". Reading carefully the new version of the manuscript I have few minor recommendations. Please find them below:

  1. Please ensure enough resolution and font size for graphs
  2. For numbers, please use one decimal where is possible (see table 1)
  3. Please better clarify how uniformity among plots was verified (e.g., water content/soil moisture sensors) 

Author Response

Response to Reviewer 2 Comments

Dear Reviewer,

Thank you very much for taking the time to review this manuscript. Please find the detailed responses below and the corresponding changes track changes in the re-submitted files.

Comments 1: Please ensure enough resolution and font size for graphs

Response 1: Thank you for pointing this out. Following your suggestion, we have adjusted the resolution and font size for most of the figures in the text. This change can be found in the revised manuscript in Figure 2, Figure 5, Figure 7, Figure S1, Figure S2, and Figure S3.

Comments 2: For numbers, please use one decimal where is possible (see table 1)

Response 2: Thank you for pointing this out. Following your suggestion, we have revised Table 1 with retaining one decimal for most numbers, but for some numbers with two decimals. For example, 0.04. If only one decimal retained, it becomes 0.0, which may affect reader’s understanding. This change can be found in the revised manuscript – page 3-4 and line 116-131.

Comments 3: Please better clarify how uniformity among plots was verified (e.g., water content/soil moisture sensors)

Response 3: Thank you for pointing this out. Since the length and width of each plot in the field are consistent, the drip irrigation placed at the center of each plot has uniform holes and flow rates in each plot. Therefore, we believed that the soil moisture content in each plot kept good uniformity.